# Amino Acid and Essential Fatty Acid in Evacuation Shelter Food in the Noto Peninsula Earthquake: Comparison with the 2024 Simultaneous National Survey in Japan

**DOI:** 10.3390/nu16234185

**Published:** 2024-12-03

**Authors:** Takamitsu Sakamoto, Hiroyo Miyata, Ayako Tsunou, Yoko Hokotachi, Satoshi Sasaki, Teruyoshi Amagai

**Affiliations:** 1Department of General Medicine, Fukuoka Tokushukai General Hospital, Fukuoka 816-0864, Japan; takamitsusakamoto@gmail.com; 2The Graduate School of Medical Sciences, Kumamoto University, Kumamoto 860-8556, Japan; 3Administration Food Sciences and Nutrition Major (Doctoral Program), Graduate School of Human Environmental Sciences, Mukogawa Women’s University, Nishinomiya 663-8558, Japan; yu111164@yahoo.co.jp (H.M.); stgrw766@ybb.ne.jp (Y.H.); 4Department of Clinical Nutrition, Kindai University Hospital, Osaka 589-8511, Japan; 5Department of Clinical Nutrition, Kitauwa Hospital, Kihoku 798-1392, Japan; 6Department of Clinical Nutrition, Takarazuka Dai-Ichi Hospital, Takarazuka 665-0832, Japan; 7Department of Social and Preventive Epidemiology, School of Public Health, The University of Tokyo, Bunkyo-ku 113-0033, Japan; stssasak@m.u-tokyo.ac.jp; 8Department of Clinical Engineering, Faculty of Health Care Sciences, Jikei University of Health Care Sciences, Osaka 532-0003, Japan

**Keywords:** disaster, digestible essential amino acid score (DIAAS), European Food Safety Authority (EFSA), half-life, N-end rule, Duplicated Combination Model

## Abstract

Background: On 1 January 2024, a 7.6 magnitude earthquake struck the Noto Peninsula. We entered the disaster area to provide relief and set up a makeshift clinic in an evacuation center to evaluate the quality and quantity of food provided there. Methods: This cross-sectional study, of mainly older adults, was conducted to analyze the amino acid and fatty acid composition of evacuation shelter meals in comparison with the results of the Japan National Survey, mainly focused on older adults. (1) We analyzed 11 evacuation foods using the “Duplicated Combination” Model and the digestible amino acid score (DIAAS) in relation to the half-life determined by the N-terminal amino acid proteins. (2) Linoleic acid (LA) and alpha-linolenic acid (ALA) levels were compared with European Food Safety Authority (EFSA) recommendations (3). The national survey of emergency food stocks in 198 hospitals and 189 social care institutions conducted in Jan 2024 was analyzed. Results: (1) DIAAS was less than 1.00 for all 11 foods provided and was considered inadequate, (2) the half-life of the protein, whose *N*-terminal valine has a half-life of 100 h, must be considered a possible deficiency when living in a shelter for more than a week, (3) LA and ALA levels were less than 40% of EFSA recommended, (4) the nationwide survey found that 80% of people have a three-day supply and data on amino acids and fatty acids were not available due to a lack of questionnaires. Conclusion: Analysis of food in evacuation shelters after the Noto Peninsula earthquake revealed the quality of amino acids involved in shelter meals using DIAAS and the lack of LA and ALA for older adults. The “Duplicated Combination” model used in this analysis could be beneficial for developing improved nutrition plans in similar future scenarios, mainly for older adults.

## 1. Introduction

On 1 January 2024, a major earthquake measuring 7.6 magnitude struck the Noto Peninsula, located on the bay coast in north-central Japan, a country known for its frequent earthquakes. The next day, the non-profit organization Tokushukai Medical Assistant Team (TMAT) entered the disaster area in Wajima City to provide relief. After completion of registration as the Disaster Medical Assistance Team (DMAT), TMAT set up a makeshift clinic in an evacuation center and began executing the subsequent tasks. This article reported that 65% of the evacuees were elderly people aged 65 and over who are vulnerable to disasters. It also reported that the food provided in evacuation centers was insufficient in energy and protein for older men and women, putting them at risk of protein-energy malnutrition (PEM) and PEM-related diseases such as infection, dehydration, and acute kidney injury. An overview of the evacuee shelters provided in the Noto Peninsula Earthquake (NPE) and the overview of food provided in the shelters have been reported elsewhere [1].

Japan is one of the most earthquake-prone countries in the world, with 221 earthquakes of magnitude 6.0 or greater recorded in the 125 years since 1901 (Figure 1), including the Great Kanto Earthquake GKE), the Great East Japan Earthquake (GEJE) and NPE, occurred on 1 September 1923 (magnitude (Mg) 8.2), 11 March 2011 (Mg 9.0) and 1 January 2024 (Mg 7.6), respectively. 

Evaluation of the quality of evacuation meals provided by hospitals and welfare facilities providing relief in disaster-stricken areas is important for reducing the incidence of disaster-related comorbidities. At this time, the government has not released any data on the amount of food provided in evacuation shelters, even though Japanese society is developing as the world’s leading super-aging society, and older adults are the most vulnerable to disaster-related comorbidity and mortality. To recognize the current status of shelter food and think about its improvement, in this study, we investigated the adequacy of the quality and quantity of food provided to NPE evacuation centers. In addition, we also aimed to compare the results with those of a nationwide survey we conducted in January 2023, in which 200 facilities were randomly selected from each of 776 government-certified Disaster Key Hospitals and Welfare facilities for disabled or older adults.

## 2. Materials and Methods

This study was designed as a cross-sectional study, mainly focused on older adults. In the present study, we examined the adequacy of the contents of the meals provided in the evacuation shelters from aspects of the composition of amino acids and fatty acids. We also compared the results with those results of a national survey of meal supplies for disasters in hospitals and welfare facilities that was planned in 2023 and conducted in January 2024. For the older adults, all results from the congregate meals and the national survey were examined.

Methods: The amino acid and fatty acid composition of 11 evacuation meals provided to evacuation shelters in Wajima City, the city most affected by the NPE, was calculated from photographs taken during medical activities by TMAT. This method has been validated [2,3]. First, the amino acid and fatty acid analysis values were calculated independently by two registered dietitians. Second, the results were compared with the digestible essential amino acid score (DIAAS), which is used to assess the protein quality of the 11 types of meal provided, and the daily intake of essential amino acids was calculated and compared with the World Health Organisation (WHO) daily intake [4]. Similarly to amino acid analysis, the fatty acid composition was also compared with the daily intake of essential fatty acids recommended by the European Food Safety Authority (EFSA) to verify validity [5]. Third, to know the quality and quantity of disaster-prepared food in hospitals and welfare facilities in Japan, the results of a nationwide survey of food stockpiled in hospitals and welfare facilities, planned for 2023 and conducted in January 2024, will be summarized. Lastly, the obtained results of the NPE and national survey will be compared.

### 2.1. Nutrient Composition Database for Foods

For calculation of the amounts of amino acids and fatty acids in the meals, foods were classified into food groups. All nutrients contained in each food group were calculated using the Standard Tables of Food Composition in Japan (8th Edition, 2024) [6]. Then, the content of all amino acids and fatty acids was calculated. In this step, when there were missing values, a method was used to fill in the missing values with similar foods to avoid underestimating the values calculated by excluding the missing values.

### 2.2. “Duplicated Combination” Model to Analyze All Possible Combinations of 3 Meals Selected from 11 Different Meals

Considering that disaster victims can freely choose three meals from 11 types of stockpiled meals, allowing overlapping in their choice, we applied the orthodox mathematical model of the “Duplicated Combination” model [7] using a programming language. It is worth noting that this method has the potential for extension to a theoretically unlimited number of meals in shelters. This novel method could be available in time for the disaster shelters to provide detailed information not only on macronutrients but also on amino acids and fatty acids individually appropriate for each victim, varying by age, sex, and co-morbidities. The details of programming commands of this method are shown in the Appendix A. The details of this novel method are written in the Appendix A of “Duplicated combination.” This model is a mathematical method for calculating combinations of multiple items, allowing for overlap. It is a method for calculating combinations of three meals per day, i.e., three items, from an unknown number of meals provided in evacuation shelters. When the total number of meals provided becomes enormous, it is important to determine which foods provide the right amount of nutrients, amino acids, and fatty acids to predict the occurrence of disaster-related comorbidities, especially for the elderly who are at high nutritional risk for sarcopenia.

### 2.3. Calculation of the Digestible Indispensable Amino Acid Score (DIAAS)

DIAAS is an index recently recommended by the FAO as the most reliable way of assessing protein quality [8]. The formula for calculating DIAAS is as follows:DIAAS (%) =100 × [(mg of digestible dietary indispensable or essential amino acid in 1 g of the dietary protein)/(mg of the reference of same dietary essential amino acid in 1 g of the reference protein)]

Here, the reference values of the essential amino acids in the denominator of the above equation for older adults are as follows: histidine 16, isoleucine 30, leucine 61, sulfur-containing amino acids (methionine and cysteine) 23, lysine 48, aromatic amino acids (phenylalanine and tyrosine) 41, threonine 25, tryptophan 6.6, and valine 40. All are expressed in the unit of mg/g of protein.

We set the cutoff value of DIAAS for older adults at 1.00. In a report comparing the quality assessment of some commercially available isolated protein sources and some commonly consumed protein-containing foods using PDCAAS and DIAAS, the results showed that scores below 1.0 were almost the same [9]. DIAAS has the advantage of being able to display scores above 1.0, whereas PDCAAS could not display scores above 1.0. Another study showed that an in vitro DIAAS of 1.00 and above is excellent, while a DIAAS between 0.75 and 0.99 is good [10]. Taking into account the age-related decline in oral and gastric digestive functions in older adults [11], we have set a cut-off value of 1.00 for the protein to be of good quality.

### 2.4. Calculation of the Quality of the Fatty Acids Involved in the Shelter Meals Compared with EFSA Recommendations

EFSA recommendations for daily requirements are proposed [5]. In order to qualify the quality and quantity of the fatty acids contained in the shelter meals, these fatty acids were compared with the EFSA recommendations and expressed in %.

### 2.5. National Survey of Disaster-Preparedness Meals

Among government-certified Disaster Key Hospitals, 776 hospitals and Welfare facilities for disabled or older adults, 200 were randomly selected. As this survey was conducted immediately after the NPE, the disaster-affected prefectures of Ishikawa, Toyama, and Niigata were excluded from the survey in order to prioritize disaster relief. A questionnaire on the quantity and quality of disaster preparedness meals was sent by post.

### 2.6. Statistical Analysis

Data are presented as mean and standard deviation. Statistical analysis was performed with SPSS version 29 (IBM, Armonk, NY, USA). The “Duplicated Combination Model” was used to randomly select three types of meals from 11 types of shelter meals, allowing for duplication. The mean and standard deviation of amino acids and fatty acids in the three selected meals were calculated. To calculate sample size in the national survey, with a 95% confidence level and 6% margin error, and population proportion and population size, we set at 50% and 800, respectively. From these settings, the sample size was calculated as 201.

### 2.7. Institutional Review Board Statement

Institutional Review Board Statement: The ethical validity of this study was reviewed by the Ethics Committee of the research institution. As a result, it was approved that it is excluded from the application of the ethical guidelines of medical and biological research works involving human subjects, and this study was approved for publication. The reference number of the Ethics Committee is gai 2404, on 5 January 2024. Moreover, the national survey was approved by the Ethics Committee of the University on 1 December 2023. The approval number is 2023-001.

## 3. Results

Three meal types were selected from the 11 meal types actually provided in NPE shelters, and overlapping combinations were allowed, resulting in 286 types. For each of these 286 dietary combinations, the daily intake of amino acids and fatty acids was calculated.

### 3.1. Daily Amino Acid Intakes for Sum of Daily Provided Three Meals

The daily intake of essential amino acids is shown (Table 1). The results of an analysis of the sufficiency rate of essential amino acids by gender, which is the ratio of the amount provided to the daily requirement of essential amino acids by gender as specified by WHO, are shown (Table 2).

This result is interpreted to mean that the amount of essential amino acids available in the shelters exceeds 100% of all essential amino acids, suggesting that there is no risk of essential amino acid deficiency in the shelters. On the other hand, because no recommended amounts of non-essential amino acids are available due to the nature of their potential producibility in humans, all data on non-essential amino acids were not analyzed. Their data are presented in Appendix A.

### 3.2. Half-Life of Ingested Protein for Disaster Victims Is Determined by N-Terminal Amino Acid

In assessing the nutritional value of foods provided in evacuation shelters in disaster areas, particularly amino acids, it is important to consider the metabolic turnover rate in the body. The metabolic turnover rate of amino acids is generally determined by the *N*-terminal amino acid of a protein, known as the “N-end rule” [12,13,14] (Tabel 3). In particular, when comparing the half-lives of essential amino acids, the half-life of valine is 100 h, and attention must be paid to valine deficiency after more than 100 h or 1 week in the shelters. In the 11 types of formula examined in this study, the ratio of the essential amino acids contained in the formula to the WHO-recommended amount of valine is greater than 1.5 for both men and women. However, the half-life of proteins with valine at the N-terminus is 100 h or about four days. Therefore, when living in a shelter for more than a week, there is a risk of developing essential valine deficiency, and attention must be paid to valine deficiency among the essential amino acids in the food provided. Conversely, in the case of deficiency of ECHS1, a key enzyme in the valine catabolic pathway, attention must also be paid to excess [15]. However, the Kapital paper did not examine special diets provided to patients with specific medical conditions, such as patients with maple syrup disease, a congenital metabolic disorder in which the patient is deficient in the enzymes that metabolize the branched-chain amino acids (BCAAs) valine, leucine and isoleucine.

### 3.3. Comparison of 11 Meals in Essential Amino Acids Using DIAAS

The results of calculating the husband’s personal DIAAS by assigning letters from A to K to each of the 11 types of food provided in the evacuation shelter during the 21 days of TMAT activities at NPE are shown below (Table 3).

As a result, the DIAAS of the 11 diets ranged from a maximum of 0.92 to a minimum of 0.6. Of these, none of the meals reached 1.00. The DIAAS cut-off is set at 100 or more/75–99 as excellent/good [5]. As the aim of the present study was to prevent malnutrition in older adults who need more leucine to prevent age-related muscle loss, we set the DIAAS cutoff value at 1.00 [9], whereas 9 out of 11 meals are considered good proteins if the cut-off value is set at 0.75. For these reasons, our result means that all 11 meals did not provide good dietary quality in terms of DIAAS, especially for older adults who are vulnerable in disaster evacuation shelters.

### 3.4. Daily Fatty Acid Intakes for 3 Meals

The mean daily intakes of the 49 fatty acids analyzed in 11 meals are shown (Table 4). The daily intakes of linoleic acid and alpha-linolenic acid as the essential fatty acids and the sum of eicosapentaenoic acid and docosahexaenoic acid were calculated. They were then compared with the EFSA daily requirement recommendations and expressed in % (Table 5). As the results compared with the EFSA recommendation, LA and ALA in the shelter meals were found to be less than 40% and could lead to essential fatty acids in the shelters. Otherwise, when the sum of EPA and DHA was calculated as the compliance rate, it was found to be more than 300%, meaning that the requirement was met. Of these, LA and ALA must be the key to the solution of the problems hidden in the shelter mail from the aspect of essential fatty acids.

### 3.5. Results of the National Survey for Disaster Prepared Meals in Hospitals and Welfare Facilities

The response rate was 54.5% (108 out of 198 hospitals) and 64.0% (121 out of 189 Welfare facilities) for government-certified Disaster Key hospitals and Welfare facilities, respectively.

Results of Hospitals: Of the hospitals that responded, 55% (59 hospitals) had more than 400 beds, and 28% (30 hospitals) had more than 20 beds. Ninety-four hospitals (87%) had a three-day supply. The reason for this is unclear. The median (25%, 75%) stockpile per person was 1956 mL (1041, 3315) water, 1400 kcal (1126, 1500) protein 40.0 g (30.0, 50.0) and salt 6.0 g (4.0, 7.9).

Results of Welfare Facilities: Of the responding facilities, 95 (78%) were nursing homes, and 26 (21%) were homes for the disabled. The number of days of storage was three days for 95 (78%), the median number of meals stored was 864 (576, 1080), and the number of meals per resident was 10 (9.00, 13.85). The median (25%, 75%) amount stockpiled per person was 480 mL (252, 860) of water, 1200 kcal (909, 1439) of energy, 36.8 g (30.0, 48.4) of protein, and 6.0 g (4.7, 8.0) of salt.

The information on the other nutrients, such as amino acids and fatty acids, was not provided by hospitals and welfare facilities due to a lack of questionnaires.

## 4. Discussion

### 4.1. Essential Amino Acids and DIAAS in Shelter Meals

On 21 September 2024, more than nine months after the NPE disaster, a Linear Precipitation hit the area exactly corresponding to the NPE, and secondary flooding also hit the NPE evacuation centers. This shows that living in an evacuation shelter for a long period of time and the meals provided in the evacuation shelter can trigger the onset of disaster-related comorbidities such as malnutrition and sarcopenia, especially in the elderly, as they are vulnerable to disasters.

Looking at Table 1 of essential amino acid intake from shelter meals in NPE, leucine is lower than the WHO recommended daily intake amount, it is a risk of developing sarcopenia for the older adults living in the shelter because of leucine role in preventing muscle loss [16]. To prevent and treat sarcopenia, the importance of leucine, among the essential amino acids that are the subject of protein quality assessment by DIAAS, has been reported [17,18,19,20]. Here, the average daily leucine intake of freely chosen three meals combination from 11 different meal types provided in the shelter was 3.4 g, which was higher than the recommended daily intake of 3 g. However, when focusing on each daily leucine intake in all possible 286 meal combinations, 65 combinations contained less than 3 g of leucine and were at risk of developing sarcopenia [21,22,23,24]. In other words, it was found that caution should be exercised with meal combinations in terms of 3 g of leucine.

Focusing on the cut-off value of DIAAS, we set it at 1.00 because of the declining tendency of older adults due to leucine deficiency mentioned above in the method session. In this discussion, if DIASS is less than 1.00 and greater than 0.75, the protein quality is considered “good.” However, we also recognize that older adults are the target of the development of sarcopenia in the shelters and are at risk for sarcopenia. Because of these concerns, we would set this from 0.75 to 1.00 to reduce the possibility of developing sarcopenia in the shelter and disaster-related morbidities.

Additionally, in recent years, it has been claimed that DIAAS indicates whether the protein is of animal or plant origin, which has traditionally been considered a simple alternative indicator of the amount of EAA. Reports on the effectiveness of sarcopenia treatment suggest that 3-hydroxy-3-methylbutyrate (HMB) [24,25], a metabolic product of leucine, may increase muscle mass, inhibit muscle loss, repair muscle and improve endurance by inhibiting protein catabolism or breakdown. The recommended dose of leucine is 3 g/day, and a maximum of 5–10% is converted to HMB. This means that a total of 30–60 g of leucine is required [24]. As this daily intake of leucine carries a risk of adverse events such as azotemia leading to acute kidney injury (AKI), even when DIAAS is high and the protein is of good quality, the provision of HBM itself as a supplement in shelters may need to be considered. In summary, from the perspective of the EAA, our results suggest that supplementation with proteins is high in DIAAS, but as leucine is insufficient even when DIAAS is high, HMB supplements may also be considered in shelter meals.

Finally, if a prolonged stay in shelters is expected, it seems important that measurement of muscle mass and strength and review of serum aminogram profiles should be considered for older adults staying in shelters for long periods after a disaster [26].

### 4.2. Fatty Acids Composition in Shelter Foods

The intake of polyunsaturated fatty acids (PUFAs) is also suggested to influence sarcopenia progression. However, although no data on *n*-3 PUFA intake in sarcopenic older adults were published, sarcopenic older adults tend to take fewer *n*-3 PUFAs compared to non-sarcopenic people, and vice versa [27,28,29]. Our analysis of the essential fatty acids in the meals provided in the evacuation shelters revealed inadequate levels of linoleic acid (LA) and alpha-linolenic acid (ALA) (Table 5). This suggests that there is a risk of essential fatty acid deficiency during long-term living in a shelter.

### 4.3. Analyses of the Results of the National Survey of Preparedness Meals in the Hospitals and Welfare Facilities

The daily energy requirement of the elderly is 2100 kcal for men and 1650 kcal for women, but the emergency food supplies in the hospital were 1400 kcal, and in the welfare facilities, they were even lower at 1200 kcal. The energy content of the stocks was 67%/85% in hospitals and 57%/73% in Welfare facilities for men/women older adults, respectively. The daily protein requirement is 60 g for men and 50 g for women, but the emergency food stocks at the hospital were 40 g, and at the welfare facilities, they were 36.8 g. Similarly, the protein supply was 67%/80% for hospitals and 61%/74% for welfare facilities. In summary, both energy and protein daily supplies from disaster food stocks in hospitals and welfare facilities were less than 75% of requirements for all except hospital food for women. It was found that the food provided in evacuation shelters over a long period of time poses a high risk of developing protein-energy malnutrition. However, it is still unclear whether the quality of amino acids and fatty acids provided by the emergency food stocks is known due to the lack of questionnaires in the national survey. This needs to be clarified in the next phase of the proposal.

### 4.4. Second Disaster Hitting the Exact Same Area Is Real

On 23 September 2024, nine months after the NPE, a linear precipitation band struck the exact same area as the NPE, causing flooding due to heavy rainfall that hit evacuation centers. This second disaster hitting and prolonged shelter stay may be a direct cause of sarcopenia in vulnerable disaster victims, similar to our experience with the Great East Japan Earthquake 13 years ago [30], leading to the occurrence of disaster-related comorbidities. The fatty acid analysis of the diet consumed in shelters is essential to prevent malnutrition due to *n*-3 fatty acid deficiency. To prevent essential fatty acid deficiency, 2 g of *n*-3 fatty acids per day is recommended [23]. It is also recommended to consume fresh fruit, vegetables, and fats or take appropriate supplements to maintain an optimal health-related quality of life [31,32]. In order to fully achieve these complex objectives, it is proposed that the disaster relief team be accompanied by nutrition experts trained in scientific observation and analysis, including knowledge of nutritional supplements.

### 4.5. Proposal of Protocol for a Nationwide Re-Survey of Amino Acids and Fatty Acids Using “Duplicated Combination” Model

We propose two novel protocols to assess the nutritional adequacy of stockpiled foods in two ways. First, we propose a novel method to calculate nutrients involved in meals in evacuation shelters in a disaster. This is a method that can be used to quickly calculate the nutritional content of the wide variety of meals served in disaster areas and select the appropriate meals to serve. This method was applied to the amino acids and fatty acids contained in each meal for the disaster stockpile meals from hospitals surveyed nationally, and the excesses and deficiencies for each age and sex were calculated using the “Dietary Reference Intakes for Japanese” (2025 version) [33], which the Ministry of Health, Labour and Welfare plans to formulate and introduce for use in April 2025, as a reference to be used in determining appropriate emergency food supplies. Second, we are using the same survey facility to send us all the photographs of the stored foods prepared in the same way as the NPE, using the double combination method to assess the appropriateness of the amino acid and fatty acid composition. This will enable us to identify any inappropriate stored foods and improve the nutritional quality of the stored foods. These proposals are, to our knowledge, the first national survey to assess the quality of disaster-prepared food.

### 4.6. Strength and Limitations

The strength of this study is that it was the first to conduct a nutritional evaluation of NPE disaster preparedness foods from the perspective of amino acid and fatty acid composition. A previous analysis of the energy and protein content of NPE pre-prepared foods showed that they were inadequate for the elderly. However, this amino acid analysis used the DIAAS, currently the most reliable indicator for assessing protein, and the results showed that the DIAAS of all 11 types of prepared meals analyses did not reach 1.00, highlighting the need to improve the protein quality of preparedness foods in the next stage. Other than DIAAS used this time, other indicators used to assess protein quality include Amino Acid Score, Biological Value (BV) [34], Net Protein Utilisation (NPU) [35], Protein Efficiency Ratio (PER) [36], and Protein Digestibility Corrected Amino Acid Score (PDIAAS), which adds digestibility to the amino acid score. Comparing DIAAS with PDIAAS, the PDCAAS uses fecal digestibility and is truncated at 100%, while the DIAAS uses ileal digestibility but is not truncated at 100% and, more importantly clinically, the DIAAS provides values for three different age groups with more recent data on human requirements. For these reasons, DIAAS is currently the most accurate score [37]. In a disaster evacuation environment, this method appears to be ideal for assessing elderly people whose digestive functions may be compromised.

The limitations of this study must also be mentioned. First, no evidence was found that older people who consumed these stored foods over a long period of time developed iron-related health problems, essential amino acid deficiency, sarcopenia, or frailty. Iron deficiency has been reported to be associated with skeletal muscle mass deficiency disorders such as sarcopenia [38]. The micronutrients have not been analyzed in this study due to a lack of data. This is an issue that needs to be addressed in the future. Second, DIAAS cut-off values specific to older people are 100 for excellent quality and 75 for good quality, which are set by FAO for the general adult population, but is 100 really appropriate, or should a DIAAS cut-off value be set specifically for older people? Further detailed consideration is needed. Third, in disaster areas where electricity is not available, it may not be possible to use the proposed double combination model with statistical software on computers in disaster environments. Therefore, it is necessary to follow the protocol foreseen in the proposal and to perform amino acid and fatty acid analysis of the disaster meals as soon as possible. Fourth, the missing values of amino acids in the food composition tables are replaced by alternative foods using the substitution method, so there is a good chance that errors have occurred. In general, the tolerance limits proposed by CODEX are limited to 20% [39,40]. However, as the amino acid content varies from food to food, we have to take into account that there are no absolute values for amino acid content. Lastly, since there are no proposals or guidelines from the Japanese government or NPOs regarding the content and nutrients of food to be provided in evacuation shelters during disasters, the results of this study are necessary to predict the prevalence of sarcopenia among the many elderly people living in evacuation shelters and the incidence rate of the disease during their stay. However, since the prevalence of sarcopenia was not observed this time, it is not possible to predict the increase in prevalence due to the provision of food. It is considered essential to conduct a survey on sarcopenia in shelters in the future.

## 5. Conclusions

Analysis of food in evacuation shelters after the Noto Peninsula earthquake revealed the quality of amino acids involved in shelter meals using DIAAS and the lack of LA and ALA for older adults. The “Duplicated Combination” model used in this analysis could be beneficial for developing improved nutrition plans in similar future scenarios, mainly for older adults.

## Figures and Tables

**Figure 1 nutrients-16-04185-f001:**
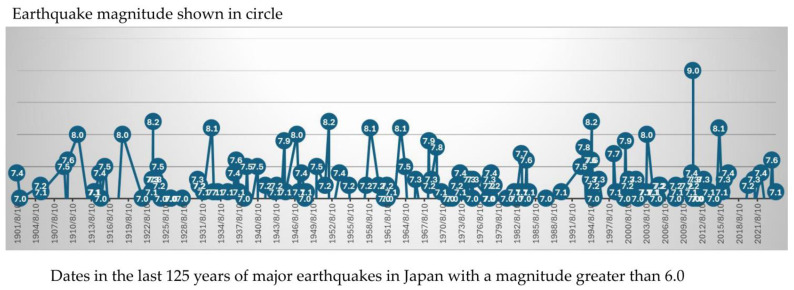
Major earthquakes of magnitude 6.0 or greater struck Japan in the last 125 years.

**Table 1 nutrients-16-04185-t001:** The daily intake of essential amino acids in the 11 foods provided to the evacuation shelter in NPE. Results of amino acid analysis of food provided for 21 days at the evacuation center run by TMAT, a non-profit organization set up to provide relief in the aftermath of the Noto Peninsula Earthquake (NPE). All essential amino acids are expressed in milligrams per one meal provided in each shelter food.

Shelter 11 Meals	His	Ile	Leu	Met	Cys	Lys	Phe	Tyr	Thr	Try	Val
	mg	mg	mg	mg	mg	mg	mg	mg	mg	mg	mg
A	283	432	785	205	172	538	471	315	383	107	492
B	297	452	790	228	163	649	467	334	413	119	507
C	603	751	1380	442	279	1172	811	667	769	235	947
D	901	910	1603	512	287	1508	885	735	921	255	1062
E	587	671	1167	394	223	1102	699	555	655	195	841
F	563	669	1224	403	228	1076	682	563	687	197	818
G	922	941	1632	618	279	1718	914	743	925	260	1142
H	409	507	938	303	190	786	548	454	520	157	643
I	651	740	1338	460	247	1261	753	626	747	214	918
J	388	489	918	292	183	774	541	441	517	153	613
K	384	473	864	277	167	766	500	413	488	146	589

Abbreviations: Cys: cysteine, His: histidine, Ile: isoleucine, Leu: leucine, Lys: lysine, Met: methionine, Phe: phenylalanine, Thr: threonine, Try: tryptophan, Tyr: tyrosine, Val: valine. A, B, C, D, E, F, G, H, I, K, J: all alphabetical letters represent 11 different meals provided in the shelter where TMAT has medical activities in NPE.

**Table 2 nutrients-16-04185-t002:** Daily intakes of essential amino acids [A], WHO recommendations [B], A/B ratios, and half-lives of proteins with each amino acid at the *N*-terminal in 11 foods provided to evacuees shelter during the NPE.

Essential Amino Acid	Average Daily Serving by 11 Meals (mg/day) [A]		WHO (mg/kg/day)	Total Amount Calculated by WHO Recommendation (mg/day) [B] *		A/B ratio		Half-Life (Hours) [9,10,11]
	Mean	SD		Male	Female	Male	Female	
His	1634.59	393.62	10	593	536	2.76	3.05	3.5
Ile	1914.91	326.63	20	1186	1072	1.61	1.79	20
Leu	3449.59	558.03	39	2312.7	2090.4	1.49	1.65	5.5
Sulfur(Met + Cys)	1811.00	315.69	15	889.5	804	2.04	2.25	30 (Met), 1.2 (Cys)
Lys	3098.47	660.14	30	1779	1608	1.74	1.93	1.3
Aroma (Phe + Tyr)	3565.82	559.10	25	1482.5	1340	2.41	2.66	1.1 (Phe), 2.8 (Tyr)
Thr	1917.64	339.68	15	889.5	804	2.16	2.39	7.2
Try	542.21	95.55	4	237.2	214.4	2.29	2.53	2.8
Val	2339.39	400.84	26	1541.8	1393.6	1.52	1.68	100

* Abbreviations: Aroma: aromatic amino acid, Cys: cysteine, His: histidine, Ile: isoleucine, Leu: leucine, Lys: lysine, Met: methionine, Phe: phenylalanine, SD: standard deviation, Sulfur: sulfur-containing, Thr: threnonine, Try: tryptophan, Tyr: tyrosine, WHO: World Health Organization recommendation, Val: valine. * Average body weight for male and female are 59.3 kg and 53.6 kg, respectively.

**Table 3 nutrients-16-04185-t003:** DIAAS on meals in evacuation shelters provided during the Noto Peninsula Earthquake. DIAAS of 11 types of meals (A-J) provided in the evacuation shelter. Only meal G had a DIAAS of 0/90, which is considered a good protein quality. The DIAAS of the other 10 types (90.9%) were all below 0.90.

Foods in NPE Shelter	G	D	I	E	F	H	K	C	J	B	A
DIAAS	0.92	0.87	0.87	0.85	0.84	0.81	0.79	0.79	0.78	0.70	0.60

Abbreviations: DIAAS: digestable indispensable amino acid score, NPE: Noto Peninsula Earthquake. A~H are the initials of the 11 types of food provided in the shelter.

**Table 4 nutrients-16-04185-t004:** The amount of each of the 49 fatty acids in 11 shelter meals.

	Mean (mg Per Meal)	SD
4:0 Butyric Acid	0	0
6:0 Hexanoic acid	0	0
7:0 Heptanoic acid	0	0
8:0 Octanoic Acid	0	0
10:0 Decanoic Acid	18.14	9.98
12:0 Lauric Acid	57.14	38.29
13:0 Tridecanoic acid	0	0
14:0 Myristic Acid	568.91	216.05
15:0 Pentadecanoic acid	41.35	17.55
15:0 ant Pentadecanoic acid	0	0
16:0 Palmitic Acid	6109.26	2227.8
16:0 iso palmitic acid	0	0
17:0 Heptadecanoic acid	126.76	45.71
17:0 ant Heptadecanoic acid	0	0
18:0 Stearic Acid	2928.69	1374.18
20:0 Arachidic Acid	101.69	48.27
22:0 Behenic acid	27.92	15.7
24:0 Lignoceric Acid	17.08	8.03
10:1 Decenoic acid	0	0
14:1 Myristoleic acid	26.76	27.06
15:1 Pentadecenoic acid	0	0
16:1 Palmitoleic acid	740.49	237.52
17:1 Heptadecenoic acid	101.43	40.3
18:1 total	13,356.86	5897.11
18:1 *n*-9 Oleic Acid	4796.7	2756.36
18:1 *n*-7 cis-Vaccenic acid	342.64	205.27
20:1 Icosenoic acid	394.86	186.42
22:1 Docosenoic acid	155.79	181.31
24:1 Tetracosenoic acid	52.95	52.4
16:2 Hexadecadienoic acid	5.55	10.91
16:3 Hexadecatrienoic acid	1.21	0.83
16:4 Hexadecatetraenoic acid	0	0
18:2 *n*-6 Linoleic Acid	4021.51	1474.02
18:3 *n*-3 α-linolenic acid	788.8	359.36
18:3 *n*-6 γ-linolenic acid	2.16	4.03
18:4 *n*-3 Octadecatetraenoic acid	70.08	82.62
20:2 *n*-6 Eicosadienoic acid	72.61	30.23
20:3 *n*-3 eicosatrienoic acid	0	0
20:3 *n*-6 eicosatrienoic acid	23.77	6.43
20:4 *n*-3 eicosatetraenoic acid	29.42	31.45
20:4 *n*-6 Arachidonic acid	108.38	50.26
20:5 *n*-3 Eicosapentaenoic acid	409.3	441.91
21:5 *n*-3 Henicosapentaenoic acid	3.43	6.73
22:2 Docosadienoic acid	0	0
22:4 *n*-6 Docosatetraenoic acid	17.07	9.13
22:5 *n*-3 Docosapentaenoic acid	92.94	80.55
22:5 *n*-6 Docosapentaenoic Acid	3	4.7
22:6 *n*-3 Docosahexaenoic acid	519.93	593.25
Unidentified substance	80.64	45.38

Abbreviations: SD: standard deviation.

**Table 5 nutrients-16-04185-t005:** Sufficiency rate of the essential fatty acids of LA, ALA, and the polyunsaturated fatty acids of EPA and DHA. The total intake of linoleic acid (LA), alpha-linolenic acid (ALA), eicosapentaenoic acid (EPA), and docosahexaenoic acid (DHA) from eating one of the 11 types of food provided in the shelter. The ratios were compared with the EFSA-recommended intakes. The results showed that intakes of both essential fatty acids, LA and ALA, were less than 40% of the recommended intake.

	Average Fatty Acid Content (g/day) [A]	EFSA Recommendation(g/day) [B] [5]	[A]/[B] Ratio (%)
Linoleic Acid (LA, 18:2 *n*-6)	4021.51	10,000	40.2
α-linolenic acid (ALA, 18:3 *n*-3)	788.8	2000	39.4
EPA + DHA	929.23	250	371.7
Eicosapentaenoic acid (EPA, 20:5 *n*-3)	409.3		
Docosahexaenoic acid (DHA, 22:6 *n*-3)	519.93		

Abbreviations: ALA: α-linolenic acid, DHA: docosahexaenoic acid, EFSA: European Food Safety Authority, EPA: ecosapentaenoic acid, LA: Linoleic Acid.

## Data Availability

The datasets generated and analyzed during the current study are available from the corresponding author upon reasonable request.

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
