# Peer review of "Amino Acid and Essential Fatty Acid in Evacuation Shelter Food in the Noto Peninsula Earthquake: Comparison with the 2024 Simultaneous National Survey in Japan"

_nutrients, 2024, doi:10.3390/nu16234185_

Round 1

Reviewer 1 Report

Comments and Suggestions for Authors

Thank you very much for allowing me to review the article titled “nutrients-3310180_Protein Quality Not Excellent and Essential Fatty Acid Deficiency in Evacuation Shelter Food Analysis in the Noto Peninsula Earthquake: Comparison with the 2024 Simultaneous National Survey in Japan” presented in the section “Geriatric Nutrition” of the Special Issue “Impact of Dietary Patterns, Nutrition, and Lifestyle on Aging and Elderly Health”.
The title of the paper is informative and reflects the approach to the content of the presented work.
In the current study, the adequacy of the meals provided in evacuation shelters is examined, focusing on the composition of amino acids and fatty acids. The results are also compared with those from a national survey on disaster meal supplies in hospitals and welfare facilities, conducted in January 2024, as part of a project planned in 2023.
The abstract is a crucial part of the article, as it contains the key information summarising the content. It is the part of the paper most readily available on all platforms, and therefore it is essential that it includes the core sections of the work within the available space. For this reason, I recommend that the authors use this space carefully and clearly and precisely state the objective of the study. There is no need to reference previous studies in the abstract.
In the methods section, the information should be organised in a logical order, specifying the population, duration, and procedures. The results appear to respond to the methodology, but there seems to be a lack of clear objectives. Typically, there should be a defined objective, with the results addressing it, while the methodology explains how the study was conducted. The conclusion should directly address the objective. For all these reasons, I suggest that the abstract be revised.
Regarding the keywords, which are essential for organising the paper and ensuring it can be found in databases, the authors should check whether the keywords used align with the MeSH (Medical Subject Headings) classification. If the terms do not correspond to this classification, the article may not be easily searchable.
The introduction sets the context of the earthquake situation, but it does not provide information on essential fatty acids or the effects of their deficiency. It also does not explain the concept of "Duplicate Combination"—information that is fundamental for contextualising the study. Relevant references are missing, and the paper does not mention the emergency food plans in place. Additionally, the hypothesis and objectives are not clearly stated. I believe the introduction needs to be revised. The authors must understand the importance of providing sufficient background information to help readers understand the study.
In the Materials and Methods section, the objective is presented, but this is misplaced. The objective should be outlined at the end of the introduction, after the hypothesis has been introduced. The information presented here, which is relevant, should have been part of the introduction. This section should focus on the study’s procedures and tools. More detail is needed on the epidemiological and statistical methods used. Additionally, approval from an ethics committee for the study is not mentioned.
The first part of the results should have been discussed in the Methods section.
The results in Tables 1 and 2 are difficult to interpret, as no comparisons are provided. The same issue applies to Tables 3 and 4. Tables 5 and 6 lack units of measurement. The data presented in the tables should be discussed in greater detail.
In the discussion, the main findings should be compared to the existing scientific literature on the topic. However, the discussion begins by reiterating the background, which is unnecessary at this stage. On the other hand, it is well structured, which aids comprehension. I am unclear about the relevance of the information presented between lines 246 and 275, as these are details that could have been included in the introduction or partially addressed in this section. I believe the study’s limitations need to be more thoroughly considered.
Finally, I do not think the conclusions should be framed in terms of a 'lesson' or critique of what may have been well-intentioned. They should, of course, propose avenues for improvement.
In conclusion, I believe this is a very interesting study, but it requires further organisation and development.
Another aspect to consider is that the paper has been submitted to a geriatrics section within a special issue focused on older adults. Therefore, the study should address this aspect more clearly throughout the work.

Author Response

We appreciate your excellent suggestion about our article.
We would like to answer your questions or suggestions one by one in the following.
Q: Your comments or questions, A: Our answers or replies

Q1, I suggest that the abstract be revised. Regarding the keywords, which are essential for organising the paper and ensuring it can be found in databases, the authors should check whether the keywords used align with the MeSH (Medical Subject Headings) classification. If the terms do not correspond to this classification, the article may not be easily searchable. 
A1, In order to cover all the data obtained in the abstract, we added a note that no data on amino acids or fatty acids were obtained from the national survey. In addition, using MeSH (Medical Subject Headings) classification, we added “Half Life” as keyword.

Q2, The introduction sets the context of the earthquake situation, but it does not provide information on essential fatty acids or the effects of their deficiency. It also does not explain the concept of "Duplicate Combination"—information that is fundamental for contextualising the study. Relevant references are missing, and the paper does not mention the emergency food plans in place. Additionally, the hypothesis and objectives are not clearly stated. I believe the introduction needs to be revised. The authors must understand the importance of providing sufficient background information to help readers understand the study. 
A2, The references of “Duplicate Combination” was added as reference 7 and 8 because it was based on a book. In addition, I added an explanation of “Duplicated Combination” in the “Duplicated Combination” Model to analyze all possible combinations of 3 meals selected from 11 different meals” in the method session as the followings:
“This model is a mathematical method for calculating combinations of multiple items, allowing for overlap. It is a method for calculating combinations of three meals per day, i.e., three items, from an unknown number of meals provided in evacuation shelters. When the total number of meals provided becomes enormous, it is important to determine which foods provide the right amount of nutrients, amino acids, and fatty acids to predict the occurrence of disaster-related comorbidities, especially for the elderly who are at high nutritional risk for sarcopenia. (Line 109 – 115)

On the emergency food plans in place, we added the sentence as the follows in the introduction:
“At this time, the government has not released any data on the amount of food provided in evacuation shelters.” (Line 60 -61) 

Q3, In the Materials and Methods section, the objective is presented, but this is misplaced. The objective should be outlined at the end of the introduction, after the hypothesis has been introduced. The information presented here, which is relevant, should have been part of the introduction. This section should focus on the study’s procedures and tools. 
A3, According to your kind suggestion, we corrected the place of the objective moving to the end of the introduction from method session.

Q4, Additionally, approval from an ethics committee for the study is not mentioned. The first part of the results should have been discussed in the Methods section. 
A4, Ethic statement was misplaced. So we corrected its place to the methods session as the followings:
“Institutional Review Board Statement
Institutional Review Board Statement: The ethical validity of this study was reviewed by the Ethics Committee of the research institution. As a result, it was approved that it is excluded from the application of the ethical guidelines of medical and biological research works involving human subjects, and this study was approved for publication. The reference number of the Ethics Committee is gai 2404, on Jan.5, 2024. Moreover, the national survey was approved by the Ethic committee of the University studied on Dec. 1, 2023. The approval number is 2023-001. ”.
(Line 158 -165)

Q5, The results in Tables 1 and 2 are difficult to interpret, as no comparisons are provided. The same issue applies to Tables 3 and 4. Tables 5 and 6 lack units of measurement.
A5, We appreciate your kind and excellent suggestions. So we have added them according to your suggestions as follows:
Table 1: The WHO Recommended Daily Intake for each essential amino acid has been added to Table 1.
Table 2: As no standard values such as WHO Recommended Daily Intake for each essential amino acid exist for non-essential amino acids, we deleted this table and moved this table to supplementary file 1, And we added the explanation as the follows in the result session:
“On the other hand, because no recommended amounts of non-essential amino acids are available due to the nature of their potential producibility in humans, all data of non-essential amino acids were not analyzed. Their data are presented in Supplementary file 1.”
 (Line 200– 203)
Table 2 (previous Table 3) :all units were added.
Table 3 (previous Table 4): GIAAS has no unit in itself.
Table 4 (previous Table 5): As we missed the unit for each fatty acid, we added it in mean (mg per meal)
Table 5 (previous Table 6) :all units were added.

Q6, The data presented in the tables should be discussed in greater detail. In the discussion, the main findings should be compared to the existing scientific literature on the topic. However, the discussion begins by reiterating the background, which is unnecessary at this stage. On the other hand, it is well structured, which aids comprehension. 
A6, According to your kind suggestion, we have added the discussion of essential amino acid deficiency in the shelter as follows in the first part of the discussion and added reference 17:
“Looking at Table 1 of essential amino acid intake from shelter meals in NPE, leucine is lower than the WHO recommended daily intake amount, it must be at risk of developing sarcopenia for the older adults living in the shelter, because of leucine role to prevent muscle loosing [17].” (Line 285 – 288)

Q7, I am unclear about the relevance of the information presented between lines 246 and 275, as these are details that could have been included in the introduction or partially addressed in this section. 
A7, By comparing the results of the national survey with the food provided in the NPE shelters, we included the results of the national survey, which happened to be conducted immediately after the NPE, to show that the problems with the food provided in the shelters were not limited to the NPE area, but could occur throughout the country. However, because the location and method of the notation could be misconstrued as an abridgement of another paper that was suddenly inserted, we have included it separately in the Methods and Results sections.

Q8, I believe the study’s limitations need to be more thoroughly considered. 
A8, According to your knowledgeable suggestion, we have added the limitation as the follows:

“Lastly, since there are no proposals or guidelines from the Japanese government or NPOs regarding the content and nutrients of food to be provided in evacuation shelters during disasters, the results of this study are necessary to predict the prevalence of sarcopenia among the many elderly people living in evacuation shelters and the incidence rate of the disease during their stay. However, since the prevalence of sarcopenia was not observed this time, it is not possible to predict the increase in prevalence due to the provision of food. It is considered essential to conduct a survey on sarcopenia in shelters in the future.”
 (Line 409 – 416)

Q9, Finally, I do not think the conclusions should be framed in terms of a 'lesson' or critique of what may have been well-intentioned. They should, of course, propose avenues for improvement. In conclusion, I believe this is a very interesting study, but it requires further organisation and development. Another aspect to consider is that the paper has been submitted to a geriatrics section within a special issue focused on older adults. Therefore, the study should address this aspect more clearly throughout the work.
A9, According to your comments, we also revised our conclusions to focus on older adults who sought refuge in shelters during disasters as the follows in the conclusion:
:” Analysis of food in evacuation shelters after the Noto Peninsula earthquake revealed that the quality of amino acids involved in shelter meals using DIAS and the lack of LA and ALA for the older adults. The "Duplicated Combination" model used in this analysis seems to be useful for investigating the quality of foods in shelters.” 
(Line 418 – 421 and that in the abstract)

Once again, we would like to thank you for your excellent comments and suggestions.
We would like to ask you to review our article again, although we know you must be too busy with your work.

Best regards

Takamitsu Sakamoto, MD
Teruyoshi Amagai, MD, PhD

Reviewer 2 Report

Comments and Suggestions for Authors

MANUSCRIPT: 3310180

TITLE: Protein Quality Not Excellent and Essential Fatty Acid Deficiency in Evacuation Shelter Food Analysis in the Noto Peninsula Earthquake: Comparison with the 2024 Simultaneous National Survey in Japan

Manuscript 3310180 “Protein Quality Not Excellent and Essential Fatty Acid Deficiency in Evacuation Shelter Food Analysis in the Noto Peninsula Earthquake: Comparison with the 2024 Simultaneous National Survey in Japan” presents an interesting study in order to evaluate the nutritional quality of meals provided in the shelters for evacuees from the earthquake that occurred in the Noto Peninsula Earthquake.

In my opinion, the topic is original and relevant to the field because there are no known studies on nutritional assessment of meals provided in disaster situations. Similar studies are not known and therefore these types of studies are relevant in order to identify possible gaps in the nutritional quality of the meals provided so that they can be corrected in future situations of catastrophic events.

This work is well structured, well planned and the research is competently carried out. In my opinion, I find that the methodology is suitable for the work carried out. No further controls should be considered.

Methodology used was adequate and completely described. The results were subject to appropriate statistical analysis and appropriately discussed and referred to the limitations of the survey-based study.

Conclusions are in accordance with the objects and results obtained.

Literature cited is adequate and most of the papers cited are from the last five years.

Regarding the manuscript, I have only minor questions for the authors to consider:

1. Table 4 – Please change the abbreviation GIAAS to DIAAS.

2. Table 1 and 2 – Please change in footnote Abbreviations to abbreviations.

3. Table 1 and 2 – Please change in footnote Abbreviations to abbreviations.

I don't have any additional comments on the tables and figures.

Author Response

We appreciate your excellent suggestion about our article.
We would like to answer your questions or suggestions one by one in the following.
Q: Your comments or questions, A: Our answers or replies
Q1, Table 4 – Please change the abbreviation GIAAS to DIAAS. 
⇒A1, Thank you for your kind pointed out. I corrected. 
Q2, Table 1 and 2 – Please change in footnote Abbreviations to abbreviations. 
⇒A2, Thank you for your kind pointed out. I corrected. 
Q3, Table 1 and 2 – Please change in footnote Abbreviations to abbreviations. 
⇒A3, Yes, I did.

Once again, we would like to thank you for your excellent comments and suggestions.
We would like to ask you to review our article again, although we know you must be too busy with your work.

Best regards
Takamitsu Sakamoto, MD
Teruyoshi Amagai, MD, PhD

Reviewer 3 Report

Comments and Suggestions for Authors

Abstract

The mention of insufficient food for older adults in evacuation shelters is important but could be clearer by specifying which nutrients are lacking, such as protein or essential fatty acids. The term "Duplicated Combination Model" should be briefly explained, as it is central to the study. While the DIAAS calculation and comparison with EFSA recommendations are noted, a bit more detail on how amino acid and fatty acid profiles were determined would help. The results are summarized well, but the sentence about DIAAS being less than 1.00 could explain why this is considered inadequate. Including details from the nationwide survey would provide a fuller picture. The conclusion could also briefly suggest ways to improve future disaster preparedness.

Introduction

The introduction explains the vulnerability of older adults but could better describe the specific nutritional challenges they face, like the risk of PEM or dehydration. It references previous reports but doesn’t include studies on the nutritional quality of disaster meals, which would help put the research into context. The study's goals are mentioned but could be stated more clearly at the beginning for better readability.

Materials and Methods

The "Duplicated Combination Model" needs clearer explanation, especially for readers who may not be familiar with it. A diagram could help. The methods for analyzing amino acids and fatty acids are explained well, but adding information about the reliability of the results would improve transparency. The manuscript doesn't mention how the sample size was calculated or what statistical methods were used-this should be included. The explanation of how the WHO and EFSA guidelines were applied could be made clearer, especially why these guidelines are relevant for older adults in a disaster setting.

Results

The tables with amino acid data are helpful, but including more summary statistics (like means and standard deviations) would make the results easier to interpret. Graphs like bar charts or heatmaps could help visualize the data better. The explanation of DIAAS values could be expanded to explain why a value under 1.00 is a concern for older adults’ nutrition. More details about the nationwide survey, such as sample size and types of facilities, would give more context and strengthen the findings.

Discussion

The discussion mentions potential iron-related health issues in older adults but doesn't provide evidence for this. A deeper explanation of the link between iron deficiency and health risks in evacuees would be useful. There’s also a good point about the DIAAS thresholds for older adults, but the discussion could go further into whether different thresholds should be used. Finally, while the "Duplicated Combination Model" is innovative, the manuscript doesn’t address how it could be used in disaster settings where resources are limited. More discussion on this would make the model more practical.

Author Response

We appreciate your excellent suggestion about our article.
We would like to answer your questions or suggestions one by one in the following.
Q: Your comments or questions, A: Our answers or replies

Q1, Abstract The mention of insufficient food for older adults in evacuation shelters is important but could be clearer by specifying which nutrients are lacking, such as protein or essential fatty acids. 
A1, According to your excellent suggestion, we corrected the abstract to show the specific nutrients lacking in the shelter meals such as essential amino acids and essential fatty acids.

Q2, The term "Duplicated Combination Model" should be briefly explained, as it is central to the study. While the DIAAS calculation and comparison with EFSA recommendations are noted, a bit more detail on how amino acid and fatty acid profiles were determined would help. 
A2, We added the explanation and its possible necessity were added the references 7 and 8. In addition, we added the explanation as follows in the Methods session:

“This model is a mathematical method for calculating combinations of multiple items, allowing for overlap. It is a method for calculating combinations of three meals per day, i.e., three items, from an unknown number of meals provided in evacuation shelters. When the total number of meals provided becomes enormous, it is important to determine which foods provide the right amount of nutrients, amino acids, and fatty acids to predict the occurrence of disaster-related comorbidities, especially for the elderly who are at high nutritional risk for sarcopenia. (Line 109 – 115)

Q3, The results are summarized well, but the sentence about DIAAS being less than 1.00 could explain why this is considered inadequate. 
Q3, As you kindly pointed out, it is essential to set the cut-off value of DIAAS, we have added the explanation in the Methods session as follows::
“We set the cutoff value of DIAAS for older adults at 1.00. In a report comparing the quality assessment of some commercially available isolated protein sources and some commonly consumed protein-containing foods using PDCAAS and DIAAS, the results showed that scores below 1.0 were almost the same [10]. DIAAS has the advantage of being able to display scores above 1.0, whereas PDCAAS could not display scores above 1.0. Another study showed that an in vitro DIAAS of 1.00 and above is excellent, while a DIAAS between 0.75 and 0.99 is good [11]. Taking into account the age-related decline in oral and gastric digestive functions in older adults [12], we have set a cut-off value of 1.00 for the protein to be of good quality. 
(Line 127 – 135)

Q4, Including details from the nationwide survey would provide a fuller picture. 
A4, We added the details of the nationwide survey in the result session as the followings:
“Results of Hospitals: Of the hospitals that responded, 55% (59 hospitals) had more than 400 beds and 28% (30 hospitals) had more than 20 beds. 94 hospitals (87%) had a 3-day supply. The reason for this is unclear. The median (25%, 75%) stockpile per person was 1,956 ml (1041, 3315) water, 1,400 kcal (1126, 1500) protein 40.0g (30.0, 50.0) and salt 6.0g (4.0, 7.9). 
Results of Welfare facilities: Of the responding facilities, 95 (78%) were nursing homes and 26 (21%) were homes for the disabled. The number of days of storage was 3 days for 95 (78%), the median number of meals stored was 864 (576, 1080) and the number of meals per resident was 10 (9.00, 13.85). The median (25%, 75%) amount stockpiled per person was 480 ml (252, 860) of water, 1200 kcal (909, 1439) of energy, 36.8 g (30.0, 48.4) of protein, and 6.0 g (4.7, 8.0) of salt. 
The information of the other nutrients, such as amino acids and fatty acids, were not provided by hospitals and welfare facilities due to a lack of questionnaires.” 
(Line 264 – 276).

Q5, The conclusion could also briefly suggest ways to improve future disaster preparedness. 
A5, According to your kind suggestion , we corrected our conclusion as the followings:
“Analysis of food in evacuation shelters after the Noto Peninsula earthquake revealed that the quality of amino acids involved in shelter meals using DIAS and the lack of LA and ALA for the older adults. The "Duplicated Combination" model used in this analysis seems to be useful for investigating the quality of meals in shelters.” 
(Line 418 -421 and corrected the abstract)

Q6, Introduction The introduction explains the vulnerability of older adults but could better describe the specific nutritional challenges they face, like the risk of PEM or dehydration. It references previous reports but doesn’t include studies on the nutritional quality of disaster meals, which would help put the research into context. 
A6, Aa the older adults are the main subjects in the evacuation shelters, we added the importance of these ages in the introduction as the followings:
“even though the Japanese society is developing as the world's leading super-ageing society, and the older adults are the most vulnerable to the disaster0related comorbidity and mortality” 
(Line 61 – 63)

Q7, The study's goals are mentioned but could be stated more clearly at the beginning for better readability. 
A7, We added the sentence in the introduction because your comment m\uts be of importance:
“To recognize the current status of shelter food and think about its improvement, in this study, we investigated the adequacy of the quality and quantity of food provided to NPE evacuation centers. In addition, we also aimed to compare the results with those of a nationwide survey we conducted in January 2023, in which 200 facilities were randomly selected from each of 776 government-certified Disaster Key Hospitals and Welfare facilities for disabled or older adults.”
(Line 64 -69)

Q8, Materials and Methods The "Duplicated Combination Model" needs clearer explanation, especially for readers who may not be familiar with it. A diagram could help. 
A8, It seems to be easy to understand the meaning of “Duplicated Combination Model”, thas the space to draw the figure is limited, we added reference s7, 8 and the explanation as the followings in the methods session:
“This model is a mathematical method for calculating combinations of multiple items, allowing for overlap. It is a method for calculating combinations of three meals per day, i.e., three items, from an unknown number of meals provided in evacuation shelters. When the total number of meals provided becomes enormous, it is important to determine which foods provide the right amount of nutrients, amino acids, and fatty acids to predict the occurrence of disaster-related comorbidities, especially for the elderly who are at high nutritional risk for sarcopenia.”
 (Line 109 – 115)

Q9, The methods for analyzing amino acids and fatty acids are explained well, but adding information about the reliability of the results would improve transparency. The manuscript doesn't mention how the sample size was calculated or what statistical methods were used-this should be included. The explanation of how the WHO and EFSA guidelines were applied could be made clearer, especially why these guidelines are relevant for older adults in a disaster setting. 
A9, For an explanation for DIAAS, we showed the equation as the followings:
DIAAS (%) =100 X [(mg of digestible dietary indispensable or essential amino acid in 1 g of the dietary protein) / (mg of the reference of same dietary essential amino acid in 1 g of the reference protein)] 

(Line 119 – 121)

For EFSA application, we added according your suggestion in the method session as the followings:
Calculation of Thequality of the fatty acids involved in the shelter meals comparing with EFSA recommendations

EFSA recommendations for daily requirements are proposed [5]. In order to qualify the quality and quantity of the fatty acids contained in the shelter meals, these fatty acids were compared with the EFSA recommendations and expressed in %.” 
(Line 137 – 142)

Moreover, we added the description for calculating the sample size as the followings:
“To calculate sample size in the national survey, confidence level, margin error, population proportion and population size we reset at 95%, 6%, 50%, and 800, respectively. From these settings, the sample size was calculated as 201.” 
(Line 155 -157)

Q10, Results The tables with amino acid data are helpful, but including more summary statistics (like means and standard deviations) would make the results easier to interpret. Graphs like bar charts or heatmaps could help visualize the data better. 
A10, we added the explanation of table 1 in the result session as the followings:
“This result is interpreted to mean that the amount of essential amino acids available in the shelters exceeds 100% of all essential amino acids, suggesting that there is no risk of essential amino acid deficiency in the shelters. On the other hand, because no recommended amounts of non-essential amino acids are available due to the nature of their potential producibility in humans, all data of non-essential amino acids were not analyzed. Their data are presented in Supplementary file 1. "
 (Line 198 – 203)

Moreover, as no standards are available for the non-essential amino acids, Table 2 was deleted and moved to supplementary file 1.

Q11, The explanation of DIAAS values could be expanded to explain why a value under 1.00 is a concern for older adults’ nutrition. 
A11, In the literature [Hammer L, 2024], protein quality is considered good with a DIAAS score of 0.75-0.99 and excellent with a score of 1.00-. On the other hand, considering the reported decrease in oral and gastrointestinal digestibility in older adults [Sánchez-García J, 2024], it was concluded that the DIAAS score for older people should be 1.0 or higher. Therefore, we added the following sentences in the methods session:

“DIAAS has the advantage of being able to display scores above 1.0, whereas PDCAAS could not display scores above 1.0. Another study showed that an in vitro DIAAS of 1.00 and above is excellent, while a DIAAS between 0.75 and 0.99 is good [11]. Taking into account the age-related decline in oral and gastric digestive functions in older adults [12], we have set a cut-off value of 1.00 for the protein to be of good quality. “
(Line 130 -135)

Q12, More details about the nationwide survey, such as sample size and types of facilities, would give more context and strengthen the findings. 
A12, The government-certified Disaster Key Hospital is less than 800. So the one of the four considered enough to overview to whole picture of the Evacuation shelter melas.

Q13, Discussion The discussion mentions potential iron-related health issues in older adults but doesn't provide evidence for this. A deeper explanation of the link between iron deficiency and health risks in evacuees would be useful. 
A13, We added the importance of iron deficiency as the followings:
“Iron deficiency has been reported to be associated with skeletal muscle mass deficiency disorders such as sarcopenia[39].”
 (Line 394 – 395) 

Q14, There’s also a good point about the DIAAS thresholds for older adults, but the discussion could go further into whether different thresholds should be used. 
A14, According to your knowledgeable comment, we added the explanation why we set the threshold of DIAAS at 1.00 as the followings in the method session as the repeating A11 and added references 7 and 8 before that:
“DIAAS has the advantage of being able to display scores above 1.0, whereas PDCAAS could not display scores above 1.0. Another study showed that an in vitro DIAAS of 1.00 and above is excellent, while a DIAAS between 0.75 and 0.99 is good [11]. Taking into account the age-related decline in oral and gastric digestive functions in older adults [12], we have set a cut-off value of 1.00 for the protein to be of good quality.”
 (Line 130 -135)

Q15, Finally, while the "Duplicated Combination Model" is innovative, the manuscript doesn’t address how it could be used in disaster settings where resources are limited. More discussion on this would make the model more practical.
A15, According to your comment, we added the explanation of “Duplicated Combinatin Model”as the followings in the method session:
“This model is a mathematical method for calculating combinations of multiple items, allowing for overlap. It is a method for calculating combinations of three meals per day, i.e., three items, from an unknown number of meals provided in evacuation shelters. When the total number of meals provided becomes enormous, it is important to determine which foods provide the right amount of nutrients, amino acids, and fatty acids to predict the occurrence of disaster-related comorbidities, especially for the elderly who are at high nutritional risk for sarcopenia.” 
(Line 109 -115)

Once again, we would like to thank you for your excellent comments and suggestions.
We would like to ask you to review our article again, although we know you must be too busy with your work.

Best regards

Takamitsu Sakamoto, MD
Teruyoshi Amagai, MD, PhD

Round 2

Reviewer 1 Report

Comments and Suggestions for Authors

I have carefully reviewed the new version of the manuscript titled “ nutrients-3310180_Protein Quality Not Excellent and Essential Fatty Acid Deficiency in Evacuation Shelter Food Analysis in the Noto Peninsula Earthquake: Comparison with the 2024 Simultaneous National Survey in Japan”. I would like to thank the authors for their efforts in enhancing the clarity and understanding of their work. 

Abstract:  From the outset, it should be clearly stated that the study focuses on older adults, specifying the sample size involved. Additionally, the study design should be included. The conclusion should emphasise that the findings could be applicable in similar future situations. These are key aspects to highlight in the abstract. 

Materials and Methods:  It is essential to clarify at the beginning that this is a cross-sectional study. The sample on which the study was conducted should be explicitly described, along with the fact that the research pertains to food for older adults. 

The results section should explicitly indicate that the focus is on the nutrition of older adults. 

The conclusion should highlight that the findings of this study could be beneficial for developing improved nutrition plans in similar future scenarios. 

Author Response

We have completed the modification according to the reviewer's comments and highlighted it in blue.

I appreciate your kind suggestions.

Best regards

Reviewer 3 Report

Comments and Suggestions for Authors

The authors have made significant revisions to clarify their methodology, strengthen the results, and provide additional context.

All reviewer comments have been addressed through specific changes to the manuscript, including appropriate referencing, additional detail and methodological clarification.